# Ice nucleation proteins self-assemble into large fibres to trigger freezing at near 0 °C

**Thomas Hansen[1], Jocelyn Lee[1], Naama Reicher[2], Gil Ovadia[3], Shuaiqi Guo[4], Wangbiao Guo[4], Jun Liu[4], Ido Braslavsky[3], Yinon Rudich[2], Peter L Davies[1]***

[1]Department of Biomedical and Molecular Sciences, Queen's University, Kingston, Canada; [2]Department of Earth and Planetary Sciences, The Weizmann Institute of Science, Rehovot, Israel; [3]The Robert H. Smith Faculty of Agriculture, Food and Environment, Institute of Biochemistry, Food Science, and Nutrition, The Hebrew University of Jerusalem, Rehovot, Israel; [4]Department of Microbial Pathogenesis, Yale University School of Medicine, New Haven, United States

*For correspondence:
peter.davies@queensu.ca

**Abstract** In nature, frost can form at a few degrees below 0 °C. However, this process requires the assembly of tens of thousands of ice-like water molecules that align together to initiate freezing at these relatively high temperatures. Water ordering on this scale is mediated by the ice nucleation proteins (INPs) of common environmental bacteria like *Pseudomonas syringae* and *Pseudomonas borealis*. However, individually, these 100 kDa proteins are too small to organize enough water molecules for frost formation, and it is not known how giant, megadalton-sized multimers, which are crucial for ice nucleation at high sub-zero temperatures, form. The ability of multimers to self-assemble was suggested when the transfer of an INP gene into *Escherichia coli* led to efficient ice nucleation. Here, we demonstrate that a positively charged subdomain at the C-terminal end of the central β-solenoid of the INP is crucial for multimerization. Truncation, relocation, or change of the charge of this subdomain caused a catastrophic loss of ice nucleation ability. Cryo-electron tomography of the recombinant *E. coli* showed that the INP multimers form fibres that are ~5 nm across and up to 200 nm long. A model of these fibres as an overlapping series of antiparallel dimers can account for all their known properties and suggests a route to making cell-free ice nucleators for biotechnological applications.

## eLife assessment

This **valuable** study provides molecular-level insights into the functional mechanism of bacterial ice-nucleating proteins, detailing electrostatic interactions in the domain architecture of multimeric assemblies. The evidence supporting the claims of the authors is **solid**, with results from protein engineering experiments, functional assays, and cryo-electron tomography, while the proposed structural model of protein self-assembly remains hypothetical. The work is of broad interest to researchers in the fields of protein structural biology, biochemistry, and biophysics, with implications in microbial ecology and atmospheric glaciation.

## Introduction

Ice crystals grow from ice embryos, which are crystalline aggregates of water molecules that spontaneously form (homogeneous nucleation) in pure H₂O at approximately −38 °C (*Hoose and Möhler, 2012*). Ice can arise in nature at much warmer temperatures because various surfaces act as stabilizers

of ice embryos (heterogeneous nucleation). Only once an ice embryo reaches a critical number of organized water molecules will it become stable enough to spontaneously grow at elevated temperatures, a process called ice nucleation (*Vali et al., 2015*). The most active heterogeneous ice nucleators are bacterial ice nucleation proteins (INPs), which can stabilize an ice embryo at temperatures as warm as −2 °C (*Lukas et al., 2022*). INP-producing bacteria are widespread in the environment where they are responsible for initiating frost (*Lindow, 1983*) and atmospheric precipitation (*Hill et al., 2014*). As such, these bacteria play a significant role in the Earth's hydrological cycle and in agricultural productivity.

As described in the literature, INPs are large proteins (up to ~150 kDa) that are thought to form multimers on the surface of the bacteria that express them (*Govindarajan and Lindow, 1988*; *Lindow et al., 1989*). AlphaFold predictions have provided some insight into the INP monomer structure (*Figure 1A*; *Forbes et al., 2022*). For the INP from *Pseudomonas borealis* (*Pb*INP) AlphaFold predicted a folded domain of ~100 residues at the N terminus followed by a flexible linker of ~50 residues, a repetitive domain composed of 65 16-residue tandem repeats, and a small 41-residue C-terminal capping structure (supported by model confidence metrics, *Figure 1—figure supplement 1*). The predicted fold of the repetitive domain agrees with some previous homology-based models in which each 16-residue repeat forms a single coil of a β-solenoid structure (*Graether and Jia, 2001*; *Garnham et al., 2011b*).

In INP sequences, most coils of the β-solenoid contain putative water-organizing motifs like Thr–Xaa–Thr (TxT) that occupy the same position in each coil to form long parallel arrays, and where Xaa is an inward pointing amino acid residue (*Figure 1B*). Shorter versions of similar arrays have convergently evolved in insects to form the ice-binding sites of several hyperactive antifreeze proteins (AFPs) (*Graether et al., 2000*; *Graham et al., 2007*; *Kristiansen et al., 2011*). These arrays are thought to organize sufficient ice-like water molecules on their surface to facilitate AFP adsorption to the ice crystal surface (*Garnham et al., 2011a*). In the much longer INP arrays, which further form multimers, the organizing effect on nearby water molecules is thought to increase to the point where ice embryos can be sufficiently stabilized to cause spontaneous growth of ice at high sub-zero temperatures. Consistent with this idea, we recently demonstrated that interrupting these water-organizing motifs decreased the ice nucleation temperature by the same amount as extensive deletions of the water-organizing coils (*Forbes et al., 2022*).

Previously, we showed that the 12 C-terminal coils lack the water-organizing motifs and that deleting these coils resulted in a near total loss of ice nucleation activity (*Forbes et al., 2022*). Interestingly, the necessity for these C-terminal coils was demonstrated by *Green and Warren, 1985* in the first publication of an INP sequence but was not further investigated. While the water-organizing coils (WO-coils) are characterized by their conserved TxT, SxT, and Y motifs, the defining feature of the C-terminal-most coils, other than the lack of these motifs, is that position 12 of the 16-residue coil is typically occupied by arginine. Thus, we refer to these non-WO-coils as R-coils. Since both coil types maintain the same predicted fold but serve different functions, we consider them subdomains of the same β-solenoid (*Kajava and Steven, 2006a*). In the WO-coils, position 12 is usually occupied by residues of the opposite charge, Asp and Glu. This charge inversion is noteworthy as it has been shown that electrostatic interactions contribute to the formation of INP multimers (*Madzharova et al., 2022*; *Lukas et al., 2020*). It has also been shown that INP activity is affected by pH, which is consistent with a role for electrostatic interactions (*Schwidetzky et al., 2021*). We and others have suggested that INPs may multimerize through salt-bridging of the side chains in these positions of the coil (*Forbes et al., 2022*; *Juurakko et al., 2022*).

Radiation inactivation analysis suggests a multimer size of 19 MDa (>100 monomers) (*Govindarajan and Lindow, 1988*). Computational estimates predict increased activity upon the assembly of up to a 5-MDa (34 INPs) multimer, which is on the same order of magnitude as that determined experimentally (*Qiu et al., 2019*). The tendency to form such large structures is one of many factors that makes these proteins difficult to work with (*Lukas et al., 2022*) and may be part of why, despite many attempts, very little is known about them at a molecular level (*Roeters et al., 2021*). The size of these structures does, however, make them amenable to size-based separation from other proteins (*Schmid et al., 1997*; *Hartmann et al., 2022*). INP multimers are also large enough to be visible using negative stain transmission electron microscopy (TEM) on enriched samples, revealing a fibril-like morphology (*Hartmann et al., 2022*; *Novikova et al., 2018*).

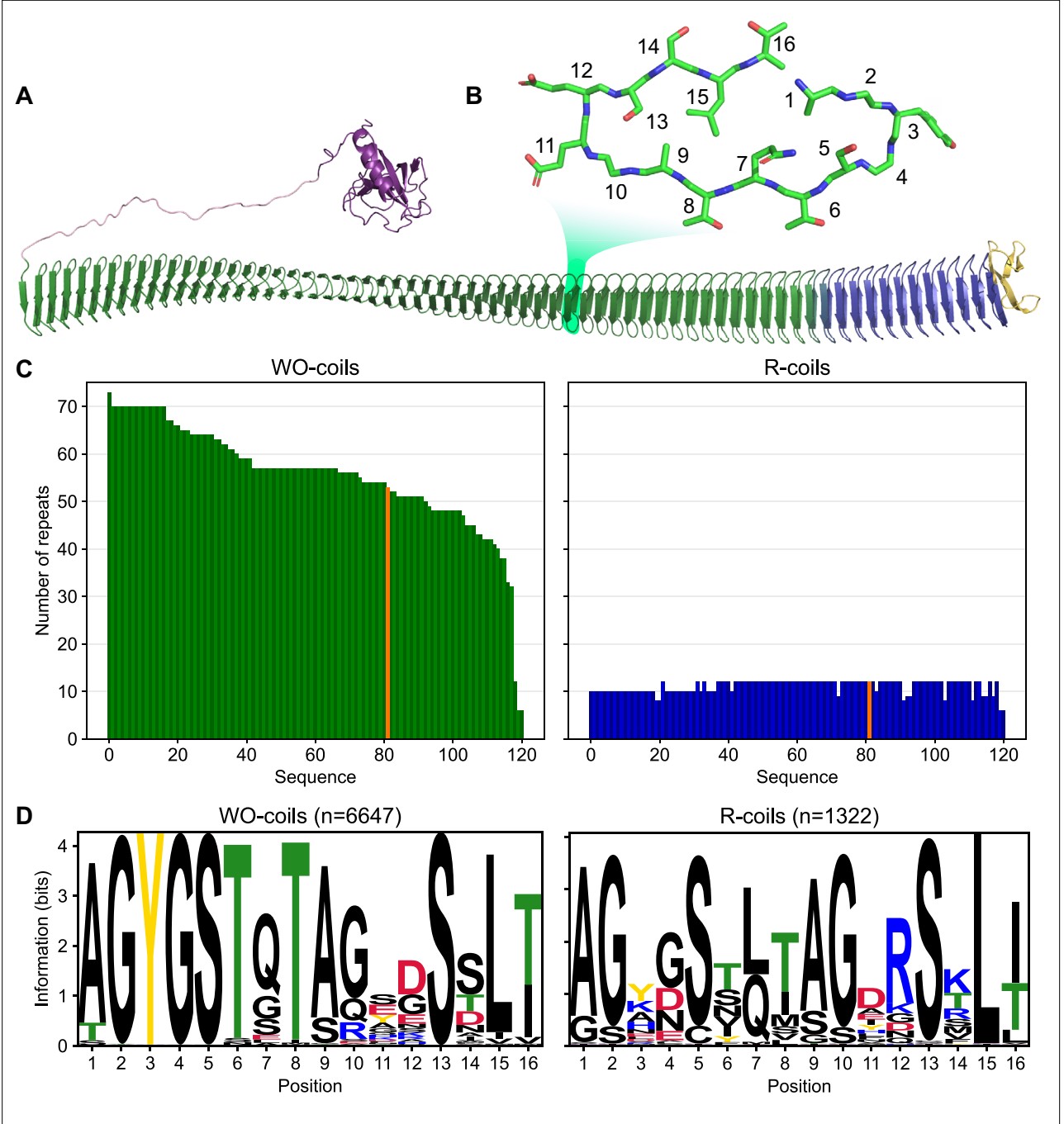

**Figure 1.** Classification of unique ice nucleation protein (INP) tandem arrays into either WO- or R-coil subdomains. (**A**) AlphaFold2 model of *Pb*INP coloured by domain. Purple: N-terminal domain, pink: flexible linker, green: water-organizing (WO) coils, blue: arginine-rich (R) coils, yellow: C-terminal cap. The inset shows a cross-section through the solenoid coil. (**B**) 16-residue tandem repeat forming one coil from the β-solenoid with positions numbered from N- to C-terminus. (**C**) The number of repeats in the WO- and R-coils for each unique sequence. *Pb*INP is included and indicated in orange. *n* = 121. (**D**) Sequence logos constructed from each 16-residue repeat present in the dataset.

The online version of this article includes the following source data and figure supplement(s) for figure 1:

**Source data 1.** Raw data table summarizing INP genes sequenced with long-read technology.

**Figure supplement 1.** AlphaFold shows high confidence in overall fold of the model.

**Figure supplement 2.** Flowchart and quality control steps in sequence selection for bioinformatic analysis.

In nature, these multimers form on the surface of bacteria, anchored to the outer membrane by the N-terminal domain (*Li et al., 2012*). When expressed recombinantly in *E. coli*, INPs have full activity, suggesting that multimers are still able to form and that they are the product of self-assembly. Remarkably, INPs with N-terminal truncations are only slightly less active, suggesting that assembly on the cell surface is not mandatory, and it can occur in the cytoplasm whether anchored to the inner surface of the plasma membrane or free in solution (*Kassmannhuber et al., 2017*; *Kassmannhuber et al., 2020*).

Here, we have studied the role of the R-coils in INPs through a series of mutations and rearrangements. Additionally, using cryo-focused-ion-beam (cryo-FIB) milling and cryo-electron tomography (cryo-ET), we have observed the fibrillar morphology of INP multimers *in situ* within cells recombinantly expressing INPs. The R-coils' length, location, and sequence are critical for INP multimerization and hence INP activity. Although we report results using *Pb*INP, the bioinformatic analysis presented here indicates that these findings are universally applicable to the INP family, including the more commonly studied InaZ from *Pseudomonas syringae*.

## Results

### Bioinformatic analysis reveals conservation in the number of R-coils across all INPs

A bioinformatic analysis of bacterial INPs was undertaken to identify their variations in size and sequence to understand what is common to all that could guide experiments to probe higher order structure and help develop a collective model of the INP multimer. In *Pb*INP, there are 53 WO-coils and 12 R-coils (*Figure 1A*), each composed of 16 residues (*Figure 1B*). To determine whether this ratio of coil types is consistent across known INPs, we analysed INP and INP-like solenoid sequences in the NCBI's non-redundant protein (nr) database. The long tandem arrays of coils in INPs make them prone to mis-assembly when using short-read DNA sequencing (*Wick et al., 2017*) so we opted to limit our dataset to sequences obtained by long-read technologies (Oxford Nanopore and Pacific Biosciences SMRT sequencing). From this bioinformatics study, it is apparent that the number of WO-coils varies considerably from over 70 coils to around 30, with a median length of 58 coils (*Figure 1C*). In contrast, the length of the R-coil region is much less variable across sequences, with 107 of 120 sequences containing either 10 or 12 R-coils (*Figure 1C*). The stark difference in length variation between the numbers of WO-coils and the R-coils supports the hypothesis that these two regions have different functions.

The differences observed between the *Pb*INP WO-coil consensus sequence and the R-coil consensus sequence, which include the loss of putative WO motifs in the former and the appearance of basic residues at position 12 in the latter, are consistent across the entire dataset (*Figure 1D*). This is apparent from the sequence logos comparing them. Also worth noting is the similarity of the sequence logos for WO- and R-coils in *Pb*INP (*Forbes et al., 2022*) with those based on 120 sequences from the database.

### Incremental replacement of R-coils with WO-coils severely diminishes ice nucleating activity

Given the remarkable conservation of the R-coil count compared to the variability of the WO-coil numbers, we measured the functional impact of shortening the R-coil region. We designed mutants in which the R-coils were incrementally replaced with WO-coils, shortening the R-coil subdomain from 12 to 10, 8, 6, 4, or 1 coil(s), while retaining the same overall length as wild-type *Pb*INP (*Figure 2A*). To avoid disrupting any potential interaction between the C-terminal cap structure and the R-coils, one R-coil was left in place to produce the 1 R-coil mutant. As previously described these constructs were tagged with GFP as an internal control for INP production, and its addition had no measured effect on ice nucleation activity (*Forbes et al., 2022*).

Ice nucleation assays were performed on intact *E. coli* expressing *Pb*INP to assess the activity of the incremental replacement mutants. In theory, replacement of R-coils by WO-coils could result in a gain of function as the water-organizing surface increased in area. However, as the number of R-coils was reduced, the nucleation temperature decreased (*Figure 2B, C*). Replacing two or four R-coils to leave ten or eight in place resulted in a slight loss of activity compared to the wild-type protein ($T_{50}$ =

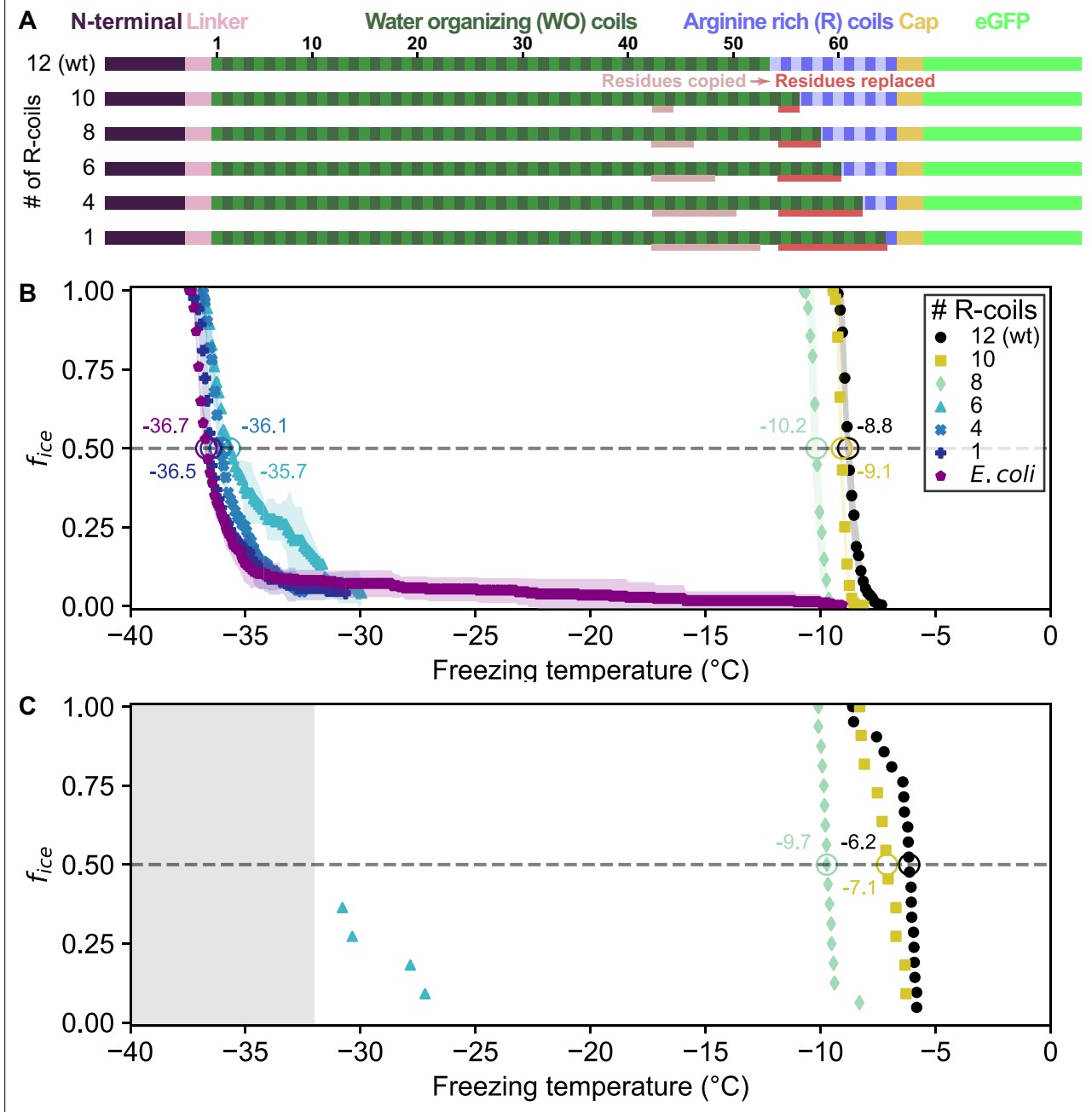

**Figure 2.** Ice nucleation activity of mutant ice nucleation proteins (INPs) in which R-coils are incrementally replaced with WO-coils. (**A**) Diagram indicating the domain map of *Pb*INP and from it, the design of the INP mutants. Sections of the WO-coils used in the replacement of the R-coils are indicated. wt: wild type. (**B**) Ice nucleation temperatures measured by WISDOM for *E. coli* cells expressing *Pb*INP and mutants with different numbers of R-coils. The temperature at which 50% of the droplets froze ($T_{50}$) is indicated with a hollow circle, and its corresponding value written nearby. The shaded region indicates standard deviation. (**C**) The same constructs assayed for ice nucleation using a nanoliter osmometer. The grey box indicates temperatures beyond the lower limit of the NLO apparatus for detecting ice nucleation in this experiment.

−9.1 and −10.2 °C, respectively, where $T_{50}$ is the temperature at which 50% of droplets have frozen, p < 0.001). Reducing the R-coil count to six dramatically decreased the activity ($T_{50}$ = −35.7 °C). The construct with only four R-coils in place showed only the slightest amount of activity, and activity was entirely lost in the construct containing only one R-coil. Evidently, small decreases in the R-coil region length produce disproportionately large decreases in activity. Halving the length of the R-coils by replacing just six coils reduced ice nucleation activity by 26.9 °C, whereas reducing the WO-coil length in half decreased the $T_{50}$ by less than 2 °C (*Forbes et al., 2022*). Since the R-coils mostly lack the

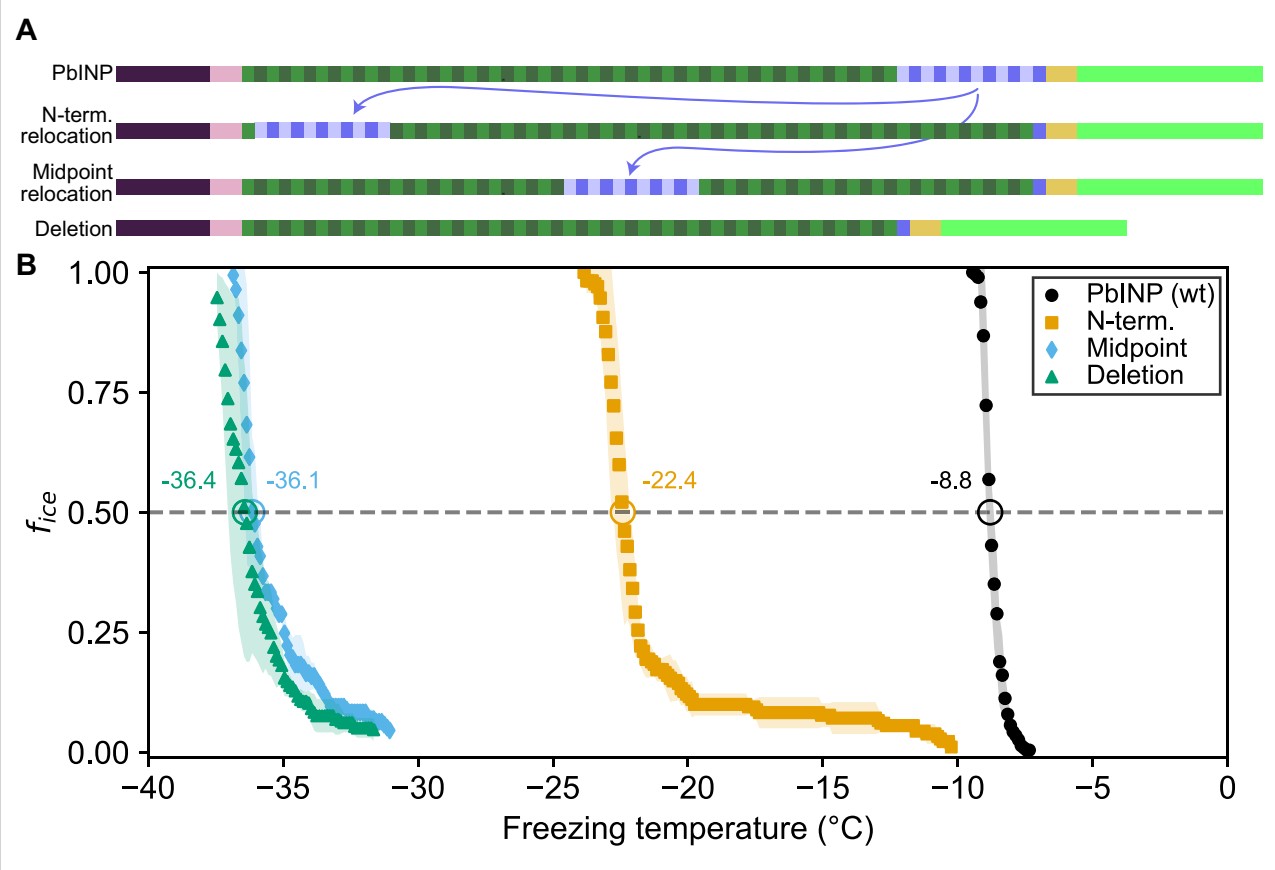

**Figure 3.** Ice nucleation activity of mutant ice nucleation proteins (INPs) with entirely relocated or deleted R-coil subdomain. (**A**) Diagram indicating the design of the constructs. Eleven of the twelve R-coil repeats were either moved within the construct or deleted. (**B**) Freezing curves with $T_{50}$ and number of unfrozen droplets indicated where applicable.

motifs required for water- organizing, we attribute the observed changes in nucleation temperature to changes in INP multimer formation.

## The location of the R-coil subdomain is crucial

In addition to its length, we investigated whether the location of the R-coil subdomain is important for ice nucleation activity. We produced constructs where 11 of the 12 R-coils were relocated to either the N-terminal end of the solenoid or the approximate midpoint of the solenoid (*Figure 3A*). As before, the C-terminal R-coil was left in place adjacent to the cap structure. We also produced an R-coil deletion construct, where the same 11 R-coils were deleted entirely from the protein.

The N-terminal relocation construct displayed markedly lower activity with a $T_{50}$ = −22.4 °C compared to wild-type *Pb*INP, and the midpoint relocation construct displayed almost no activity ($T_{50}$ = −36.1 °C), and was indistinguishable in activity from the construct where the R-coils were deleted (*Figure 3B*).

## Targeted mutations reveal that positively charged residues are important for R-coil function

Having established the importance of R-coil position and length for high activity, we next investigated the features of this subdomain that are required for its activity. Looking at the charge distribution along the solenoid from N terminus to C terminus (*Figure 4A*), we noted a switch at the start of the R-coils from an abundance of acidic residues to their replacement by basic residues. To probe the significance of this observation, we mutated all basic residues (R/K/H) in the R-coils to match those found in the same repeat positions of the WO-coils (D/G/E for positions 11 and 12, and S for position 14) (*Figure 1D*). In total, 17 basic residues – 10 arginine (R), 4 lysine (K), and 3 histidine (H) – were

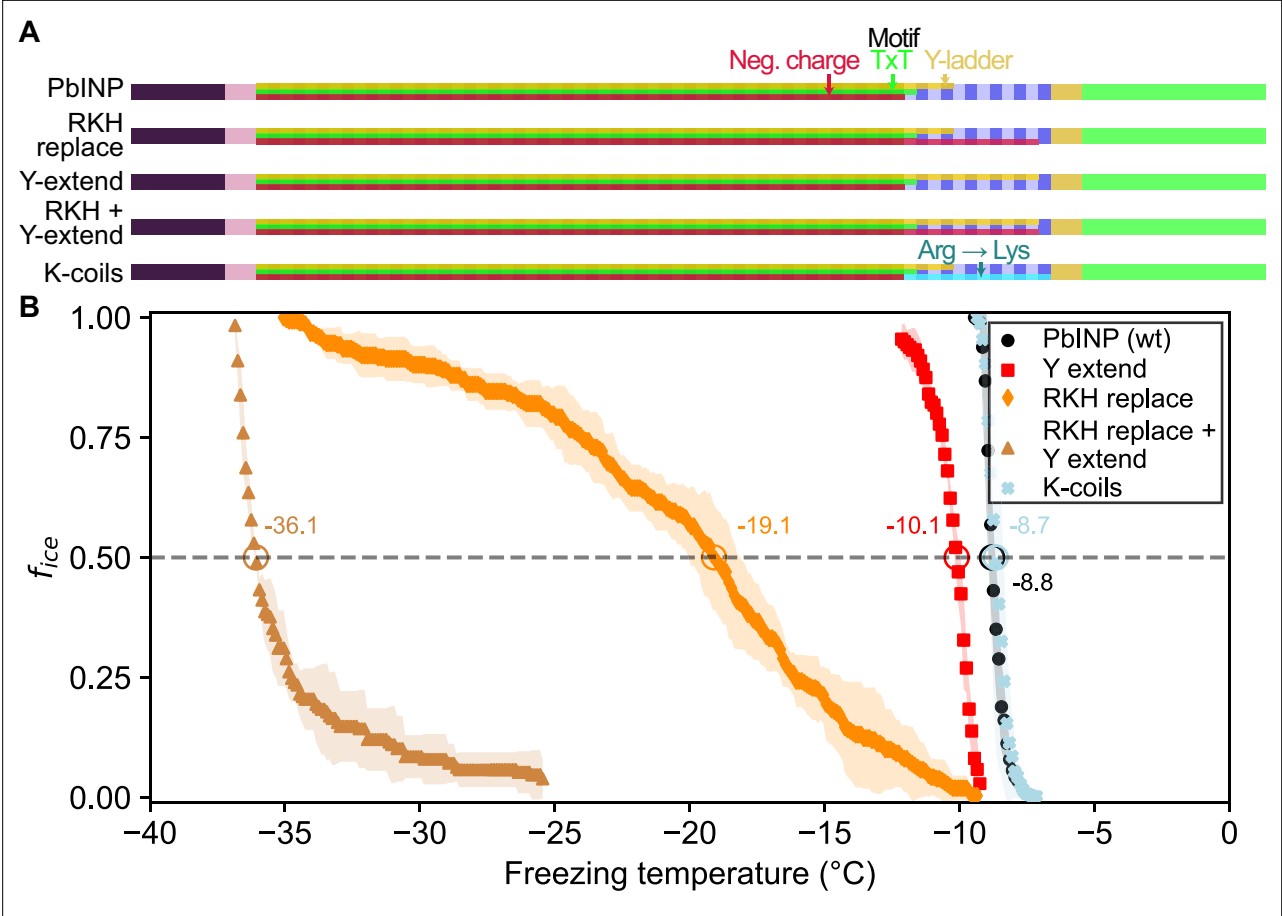

**Figure 4.** Site-specific mutagenesis of noteworthy motifs in the R-coil subdomain. (**A**) Diagram indicating the design of the constructs. Translucent bars indicate continuity of three conserved motifs along the length of the central repetitive domain. Neg. charge: negative residues present in positions 11, 12, and 14 of the repeat. TxT: Thr–Xaa–Thr motif in positions 6–8 of the repeat. Y-ladder: an entirely conserved Tyr in position 3 of the motif. Three mutants were created in which these motifs were extended into the R-coils. K-coils: all arginines in the R-coils were replaced with lysines. (**B**) Freezing curves with $T_{50}$ and number of unfrozen droplets indicated where applicable.

replaced in the R-coils to generate the RKH replacement mutant. The side chains at these positions are predicted by the AlphaFold model to point outward from the solenoid, so these mutations are unlikely to compromise the stability of the solenoid core.

There was a 10.3 °C drop in $T_{50}$ from wild-type activity after RKH replacement (*Figure 4B*) ($T_{50}$ = −19.1 °C). Although this is a large decrease in activity, it was not as deleterious as the relocation or deletion mutations (*Figure 3*). The prominent, entirely conserved tyrosine in position 3 of the WO-coils is only present in the first three R-coils and is missing from the following nine coils, making it another candidate for mutation. Upon extending this 'tyrosine ladder' through the R-coils (*Figure 4A*), there was a 1.3 °C loss in activity. However, when combining the RKH replacement with the tyrosine ladder extension, an almost total loss of activity was observed ($T_{50}$ = −36.1 °C on WISDOM) (*Figure 4B*).

In the final mutated *Pb*INP construct in this series, all arginines in the R-coils were replaced by lysines (K-coils). This mutant nucleated ice formation at essentially the same temperature as the wild type (*Figure 4B*) (p = 0.89), suggesting that positive charges in these locations are more important than side chain geometry.

## Droplet freezing assays show recombinant cell lysate supernatant has ice nucleation activity that is affected by pH

The experiments described above were performed using whole recombinant bacteria rather than extracted INPs. In *E. coli*, the vast majority of the expressed INP is intracellular (*Kassmannhuber et al., 2017*). Indeed, with our GFP-tagged constructs, we observe intense green fluorescence in the

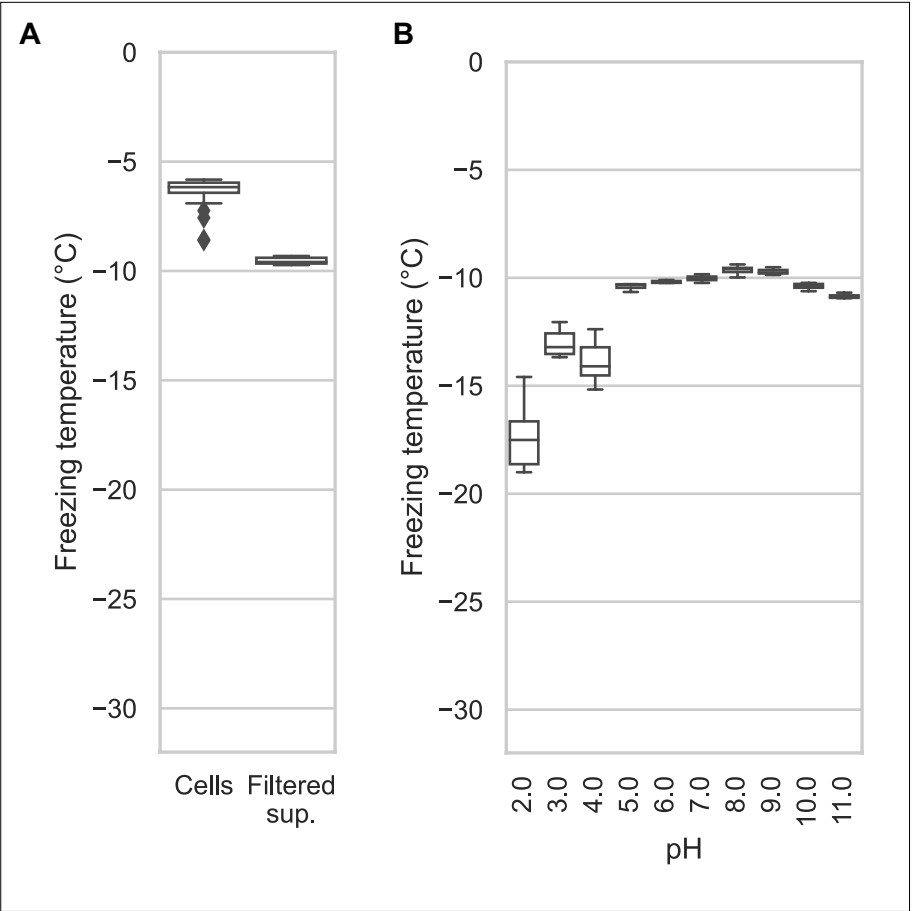

**Figure 5.** Assaying the activity of INP-containing supernatants across the pH scale. (**A**) A comparison of the nucleation temperatures of *Pb*INP when assayed using intact *E. coli* cells and when assayed with filtered supernatant. (**B**) A box and whisker plot showing the nucleation temperatures of filtered supernatant containing *Pb*INP under different pH conditions. Boxes and bars indicate quartiles, with medians indicated by a centre line. Outliers are indicated by diamonds.

cytoplasm. To see how important electrostatic interactions were in the multimerization of *Pb*INP as reflected by its ice nucleation activity, it was necessary to lyse the *E. coli* to change the pH surrounding the INP multimers. After centrifuging the sonicate to remove cell debris and passing the supernatant through a 0.2-µm filter to remove any unbroken cells, the extracts were tested to see how ice nucleation activity is affected by pH between 2.0 and 11.0. The activity of the filtered supernatant was only a few degrees lower than that of whole bacteria ($T_{50}$ = −9.6 °C) (*Figure 5A*), which agrees with the results of *Kassmannhuber et al., 2020*. This indicates that large INP structures are present within the bacterial cytoplasm.

The effect of pH on Snomax activity has been previously reported (*Lukas et al., 2020*). However, Snomax is comprised of freeze-dried *P. syringae* cells in which the INPs are thought to be membrane bound. Our assays on bacterial lysate tested free, cytoplasmic *Pb*INP complexes, producing a similar trend regarding the effect of pH but with somewhat greater loss of activity on the lower end of the optimal range (*Figure 5B*). Ice nucleation activity decreased by a few degrees below pH 5.0, and by ~8 °C at pH 2.0. The loss of activity in the alkaline buffers up to pH 11.0 was minimal. Similar to the findings of *Chao et al., 1994*, we did not observe a major change in activity (i.e. $\Delta T_{50}$ > 10 °C) even at the extremes of pH 2.0 and 11.0, suggesting that the mechanism of ice nucleation is not pH-dependent.

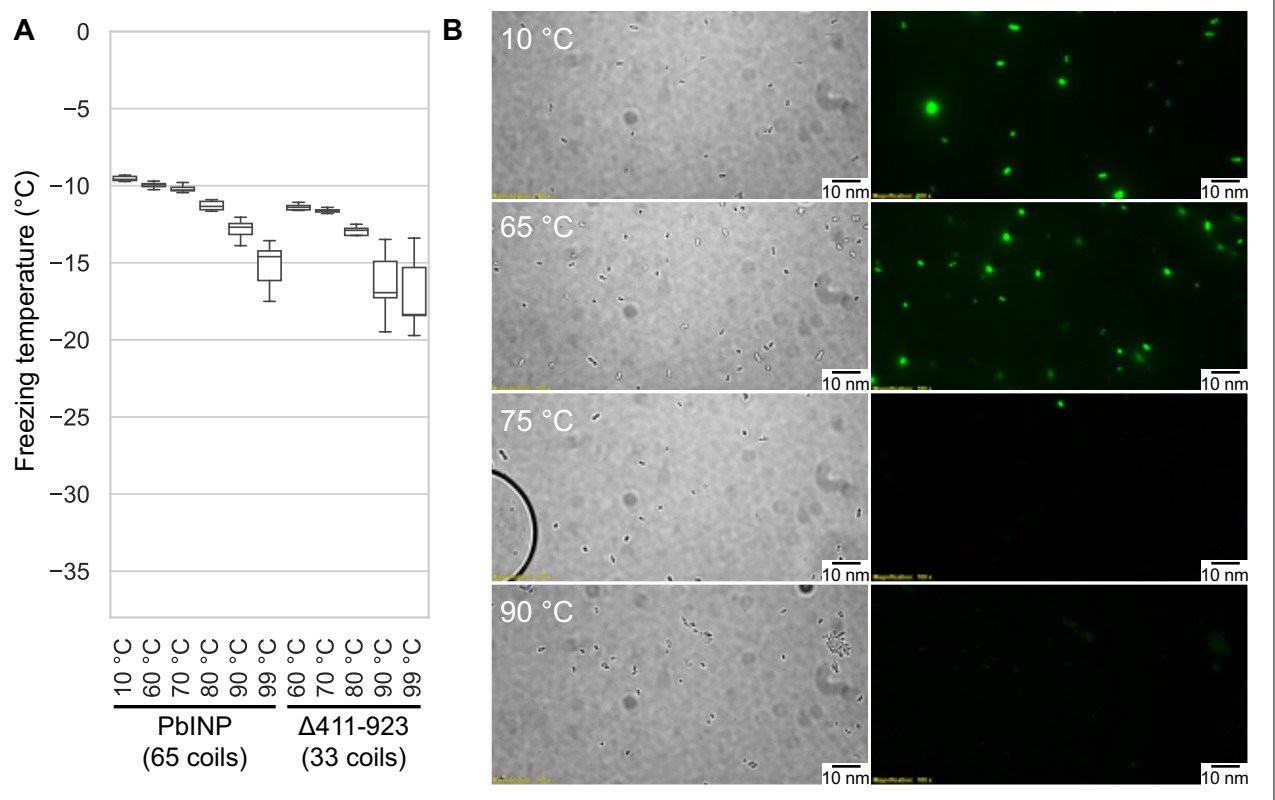

**Figure 6.** Assaying the heat stability of INPs. (**A**) Measured freezing temperatures of heat-treated droplets containing either *Pb*INP or *Pb*INP with the first 32 repeats (counting from the N-terminal end) deleted. Wild-type *Pb*INP freezing without heat treatment (kept at roughly 10 °C) is indicated on the left. (**B**) Fluorescent microscopy images of recombinant *E. coli* cells expressing *Pb*INP tagged with GFP viewed under bright-field (BF) or fluorescent excitatory (GFP) light. Representative images are shown (*n* = 3). Note: cells that retain their fluorescence after 75 °C treatment are rarely observed.

## INP activity is remarkably heat resistant

Having access to lysate also provided an opportunity to examine the heat stability of the INP complexes. The filtered lysate was heat treated to 60, 70, 80, 90, or 99 °C for 10 min in sealed tubes before being chilled and assayed for ice nucleation activity (*Figure 6A*). The activity of the 60 °C sample ($T_{50}$ = −9.9 °C) was nearly identical to the non-treated wild-type control ($T_{50}$ = −9.6 °C), and the 70 °C sample only displayed a minor loss of activity ($T_{50}$ = −10.2 °C). From 80 to 99 °C the activity incrementally decreased ($T_{50}$ = −11.3, −12.7, and −14.6 °C, respectively), but the activity loss never exceeded 6 °C. Indeed, the heat resistance of the INP complex is remarkable. The C-terminal GFP tag provided an internal control for the effectiveness of heat treatment, as GFP denatures at around 73 °C (*Melnik et al., 2011*). The green colour of the bacteria was robust at 65 °C and with very few exceptions, gone at 75 °C (*Figure 6B*). There was no fluorescence at 90 °C.

To assess what role the WO-coils play in multimer stability, we also assayed the lysate of a construct from our previous study in which repeats 16–47 (residues 411–923) of the solenoid had been deleted, leaving 32 coils (*Forbes et al., 2022*). The overall freezing profile remained the same, indicating that this construct is also extremely resistant to heat denaturation, with each temperature sample freezing at slightly lower temperatures than their full-length counterparts (*Figure 6A*). While the Δ411–923 construct had slightly lower overall activity, its heat resistance was not affected by the truncation, suggesting that the R-coil and C-terminal cap subdomains are mainly responsible for multimer stability.

## The β-solenoid of INPs is stabilized by a capping structure at the C terminus, but not at the N terminus

There is a clear C-terminal capping structure in the AlphaFold model (*Figure 1A*), but a possible N-terminal cap was more nebulous. Most protein solenoids are N- and/or C-terminally capped to help maintain the fold and/or prevent end-to-end associations (*Kajava and Steven, 2006b*). Looking at

the N-terminal sequence, we tested if any part of the extended linker region serves as an N-terminal capping motif. To investigate this, we made a series of incremental N-terminal deletions starting at residues Asp150 (Truncation 1), Gln159 (Truncation 2), and Gln175 (Truncation 3) (*Figure 7A*). Truncation 1 lacked most of the N-terminal domain, leaving the last few residues of the unstructured linker. Truncation 2 removed those linker residues so that the putative cap (a single β-strand) was located at the very N-terminal end of the protein. Truncation 3 removed the β-strand along with the rest of the first coil of the solenoid. When tested, there was no difference between the activities of the three truncations and the wild type (p = 0.82) (*Figure 7D*). This result is in line with those from *Kassmann-huber et al., 2020*, which showed that deletion of the N-terminal domain does not significantly affect ice nucleation activity.

Previously, we demonstrated that the C-terminal cap is essential for ice nucleation activity (*Forbes et al., 2022*). Bioinformatic analysis showed a high degree of conservation in the C-terminal cap residues (*Figure 7B*). Rather than deleting the cap, we made targeted mutations: F1204D, D1208L, and Y1230D, to disrupt the structure predicted by AlphaFold. Residues for mutations were chosen based on the putative key roles of those residues in the AlphaFold model. For an enhanced effect of the mutations hydrophobic residues were replaced with charged ones and vice versa. F1204 sits atop the final R-coil to cover its hydrophobic core. D1208 helps to maintain a tight loop through strategic hydrogen bonds, and Y1230 fills a gap in the surface of the cap (*Figure 7C*). When comparing these selections to the aligned C-terminal cap sequences, we see that all three residues are highly conserved. The resulting triple mutant displayed greatly reduced activity ($T_{50}$ = −27.8 °C), which helps validate the AlphaFold-predicted structure of the cap and its importance to the stability of the solenoid it covers.

## Cryo-ET reveals INPs multimers form bundled fibres in recombinant cells

The idea that INPs must assemble into larger structures to be effective at ice nucleation has persisted since their discovery (*Govindarajan and Lindow, 1988*). In the interim the resolving power of cryo-EM has immensely improved. Here, we elected to use cryo-ET to view the INP multimers *in situ* and avoid any perturbation of their superstructure during isolation. *E. coli* cells recombinantly overexpressing INPs were plunge frozen and milled into ~150 nm thick lamella using cryo-FIB (*Figure 8A*). Grids containing lamellae were transferred into either a 200- or a 300-kV transmission electron microscope for imaging under cryogenic conditions. Many *E. coli* cells were observed within the low-magnification cryo-TEM overview image of the lamella (*Figure 8B*). Tilt series were collected near individual *E. coli* cells, and 3D tomograms were reconstructed to reveal cellular and extracellular features. Strikingly, *E. coli* cells overproducing wild-type INPs appear to be lysed after 3 days of cold acclimation at 4 °C and contain clusters of fibres in the cytoplasm (*Figure 8C–E*, tomograms in *Figure 8—videos 1 and 2*). Individual fibres are up to a few hundred nanometres in length but only a few nanometres in width. Intriguingly, these fibre clusters were not observed in *E. coli* that overexpress INP mutants lacking R-coils, and the cell envelopes stay integral after being cold acclimated over the same period as those of wild-type INP-producing *E. coli* (*Figure 8—figure supplement 1*).

## Discussion

Previously, we showed that the *Pb*INP solenoid domain is made up of two subdomains: the larger N-terminal region of WO-coils accounting for 80–90% of the total length; and the smaller C-terminal R-coil region accounting for the remaining 10–20% (*Forbes et al., 2022*). The length of the WO-coil region and the continuity of the water-organizing motifs were shown to directly affect ice nucleation temperature. Although the R-coil region lacks water-organizing motifs, its presence was critical for ice nucleation activity, which led us to propose a key role for this region in INP multimer formation. Here, we have characterized the R-coil subdomain in terms of the attributes it needs to support INP multimerization and have shown by cryo-ET the first *in situ* view of what these multimers look like. In addition, we have advanced a working model for the INP multimer structure that is compatible with all of the known INP properties.

In the aforementioned work, we showed that removal of up to half of the *Pb*INP solenoid (reducing the number of WO-coils from 53 to 21) only dropped the ice nucleation activity by ~2 °C. Here, we

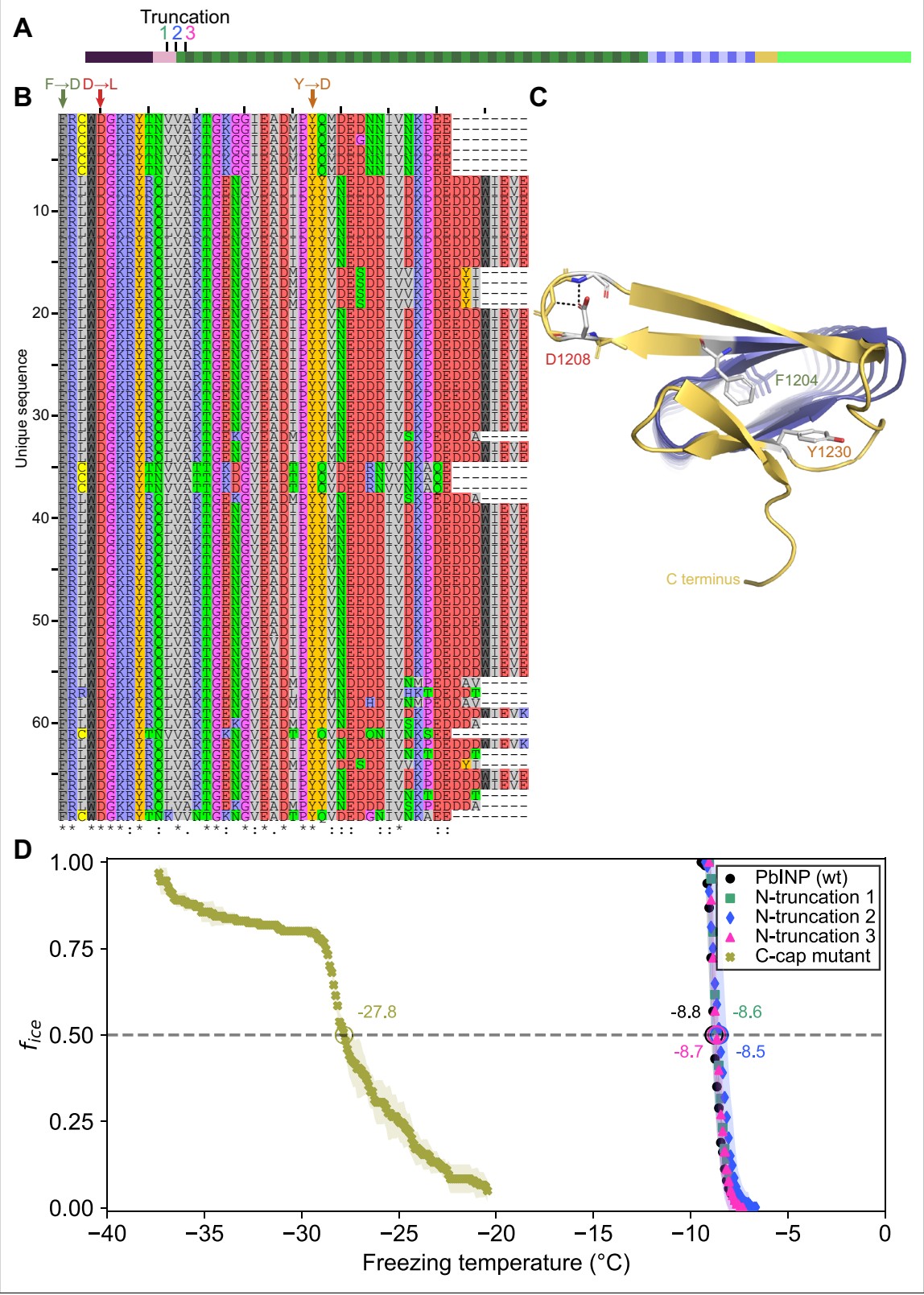

**Figure 7.** Investigating putative INP cap structures through a series of mutant proteins. (**A**) Sites of N-terminal truncations to *Pb*INP, indicating the location of the starting residue in the shortened construct. (**B**) Alignment of representative ice nucleation protein (INP) C-terminal domains from the genus *Pseudomonas*. Mutated residues and their one-letter codes are indicated above. Symbols at the bottom indicate consensus (* for fully conserved, : for conservation of strongly similar chemical properties, . for conservation of weakly similar chemical properties). (**C**) Predicted location of mutated

*Figure 7 continued on next page*

Figure 7 continued

residues in the *Pb*INP C-terminal cap with side chains shown and predicted H-bonds for D1208 shown as dashed lines. (**D**) The ice nucleation curves for the N- and C-terminal cap mutants.

have confirmed this tolerance of WO-coil count variation through bioinformatic analysis of natural INPs. The majority of bacterial INPs have WO-coil counts between 30 and 70. *Pb*INP is average in this respect with 53 WO-coils. It seems counterintuitive that these bacteria have not been uniformly selected for the highest WO-coil count, which might give them an advantage in causing frost damage to plants at the highest possible temperature (*Lindow et al., 1982*). However, it is clear that INPs are not functioning as monomers but rather as large multimers, so any loss of water-organizing surface can potentially be compensated for by simply adding more monomers to the multimer.

The ability to form superstructures is a key property of INPs and centres on the R-coil subdomain. This was shown here in the same bioinformatic analysis where there is remarkably little variation to the R-coil length of 10–12 coils. The importance of a minimal R-coil region length is supported by

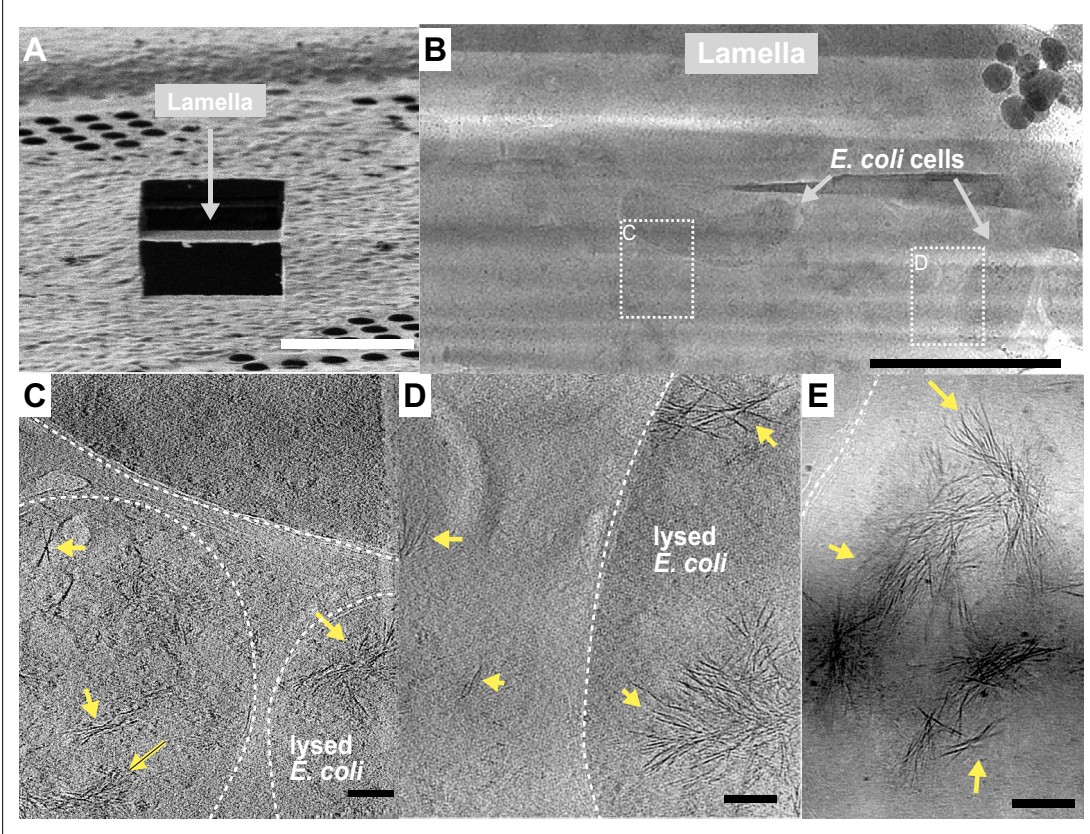

**Figure 8.** Fibrous bundles observed by cryo-focused-ion-beam (cryo-FIB) and cryo-electron tomography (cryo-ET) in in *E. coli* cells expressing ice nucleation protein (INP). (**A**) Ion-beam image of a thin lamella containing *E. coli* cells expressing INP obtained from cryo-FIB milling. (**B**) Zoomed-in view of a cryo-transmission electron microscopy (TEM) image of the lamella in (**A**). Boxes with dashed lines indicate areas where tilt series were collected. (**C, D**) Snapshots from 3D cryo-tomograms reconstructed from tilt series collected in the boxed regions in (**B**) showing striking fibrous bundles (yellow arrowheads). The *E. coli* cell envelopes are indicated with thick dash lines. (**E**) Further examples of the fibrous bundles produced by INP-expressing *E. coli*. Size markers in (**A**) is 10 μm, in (**B**) is 2 μm, and in (**C**), (**D**), and (**E**) are 100 nm, respectively.

The online version of this article includes the following video and figure supplement(s) for figure 8:

**Figure supplement 1.** *E. coli* expressing ice nucleation protein (INP) mutant lacking R-coils show no fibre clusters as observed in those cells overexpressing wild-type INP.

**Figure 8—video 1.** Tomogram of an *E. coli* cell expressing an ice nucleation protein (INP).
https://elifesciences.org/articles/91976/figures#fig8video1

**Figure 8—video 2.** Tomogram of an *E. coli* cell expressing an ice nucleation protein (INP).
https://elifesciences.org/articles/91976/figures#fig8video2

experiments. Whereas over 30 of the WO-coils can be removed with slight loss of activity, when six of the 12 *Pb*INP R-coils were replaced by WO-coils there was a catastrophic loss of ice nucleation activity, and no activity at all with further shortening of the R-coils. We postulate that at least eight R-coils are required for efficient multimer formation and that the ice nucleation activity of a monomer is inconsequential in the natural environment.

In the absence of detailed structural information, we have probed the properties of the multimers to help develop feasible models for their structure and assembly. The location of the R-coils at the C-terminal end of the solenoid next to the highly conserved cap structure is critical, as they do not function in the middle of the WO-coil region, and only poorly at the N-terminal end. These R-coils have a strong positive charge from the Arg and Lys residues, whereas the WO-coils are negatively charged, and their interaction potentially provides an electrostatic component to the fibre assembly. As expected, changing the charge on the R-coils from positive to negative caused some loss of ice nucleation activity (~9 °C), consistent with charge repulsion between these two solenoid regions weakening the multimer structure. In wild-type INPs, the negative charges of the WO-coils are consistent throughout their length, which offers no clue as to where on the WO-coils the R-coils might interact. One possible advantage of this uniformity is that multimer assembly could still happen if the WO-coil length is appreciably shortened, as it can be in nature and by experimentation (*Forbes et al., 2022*).

The minimal effects of pH change on native INP activity are reminiscent of the insensitivity of antifreeze activity to pH (*Chao et al., 1994*; *Liu et al., 2016*). The ice-binding sites of AFPs are typically devoid of charged residues and there should be no effect of pH on the ability of these sites to organize ice-like waters. The same can be said for the water-organizing motifs in INPs. We noted the extraordinary heat stability of INP multimers. Even after heating to 99 °C for 10 min, the bacterial extracts only lost 5 °C of ice nucleation activity, whereas the heat-stable internal GFP control was

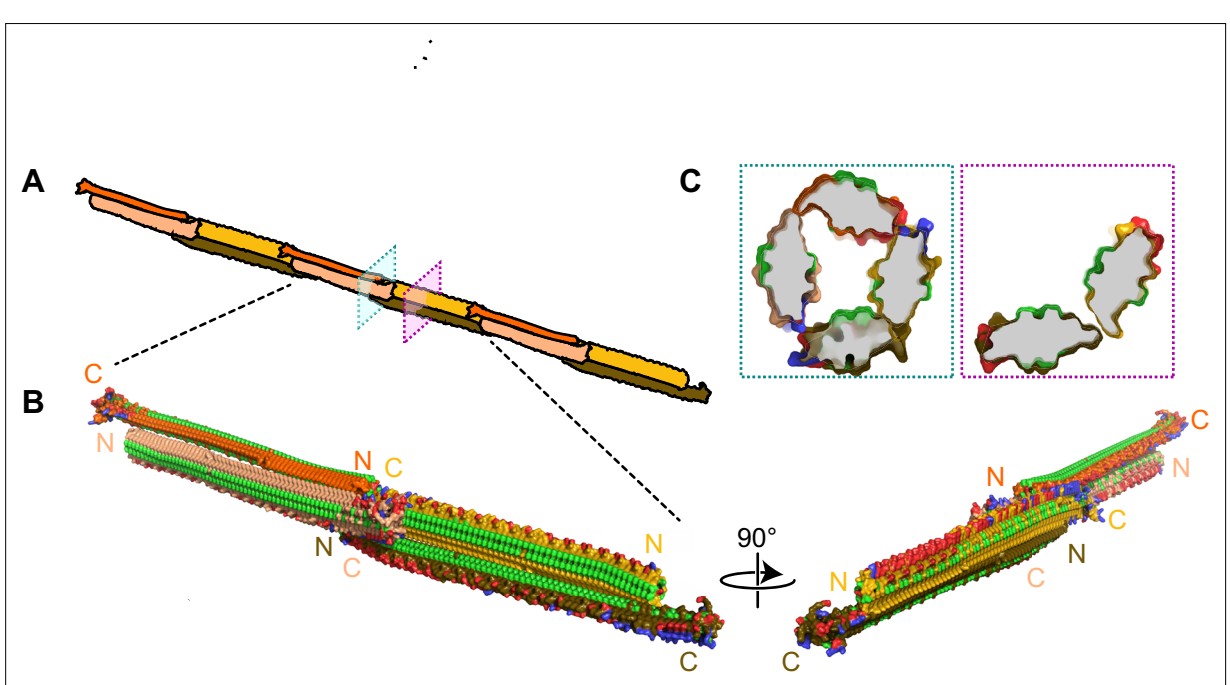

**Figure 9.** Filamentous multimer model for bacterial ice nucleation proteins (INPs). (**A**) A possible assembly of INP solenoids to form long fibres composed of antiparallel INP dimers (indicated by orange and yellow pairs). (**B**) Dimers are formed along the tyrosine ladder, a previously proposed dimerization interface. They are joined end to end by forming electrostatic interactions between negatively (red) and positively (blue) charged surfaces. All threonines are coloured light green, displaying the arrays of TxT WO motifs. The termini of the INP solenoids are labeled N and C and coloured to match panel A. This illustration uses a manually flattened AlphaFold model of *Pb*INP. (**C**) Cross-sections of the model at positions indicated in (A). Monomers are rotated approximately 90° to each other and dimerized along their tyrosine ladders (purple). Towards their termini, a pair of dimers can be matched by oppositely charged electrostatic surfaces (teal).

The online version of this article includes the following video for figure 9:

**Figure 9—video 1.** Apparent steric clash when proposed ice nucleation protein (INP) dimers are aligned in a parallel orientation.

https://elifesciences.org/articles/91976/figures#fig9video1

denatured at 75 °C. We cannot rule out the possibility that the INP multimers were also denatured by heat treatment but could reassemble on cooling.

## Working model of the INP multimer

The fundamental unit of the INP multimer in this hypothetical model is a dimer (*Figure 9*). The dimerization interface involves an interaction of the stacked tyrosine ladders from the two INP monomers as previously suggested (*Garnham et al., 2011b*; *Hartmann et al., 2022*). However, in this model the INPs are aligned antiparallel to each other (*Figure 9B*). This orientation is more likely than a parallel alignment since the R-coils and C-terminal cap structure appear to clash when modelled parallel to each other (*Figure 9—video 1*). The antiparallel dimer would not be a rigid, flat sheet but could hinge at the tyrosine ladder. Another advantage of the antiparallel arrangement is that the two dimer termini are identical, allowing end-to-end linking to form a long fibre.

The end-to-end dimer associations involve electrostatic interactions between the basic side of the R-coils and the acidic side of the WO-coils. If these interactions can also form with the proteins at an approximate right angle, it should be possible for end-linked dimers to form a compact fibre (*Figure 9B*) with a diameter close to that seen by cryo-ET (*Figure 8*). The antiparallel arrangement of the dimers gives a sidedness to the multimer where TxT motifs (light green) face outwards and inwards in an alternating pattern with SxT motifs (on the underside) in the opposite phase (*Figure 9B*). Cross-sectional views of the INP fibre (*Figure 9C*) show the interactions between the negatively (red) and positively (blue) charged regions where the dimers overlap to form a ring of four solenoids, and here and elsewhere the interaction of the two monomers through the tyrosine ladder pairing.

## Working model of the INP multimer is consistent with the properties of INPs and their multimers

We previously showed that the length of the WO-coil region can be shortened by ~60% with only a few degrees Celsius decrease in ice nucleation temperature (*Forbes et al., 2022*). The working model can accommodate these huge deletions simply by closing the gap between the dimers. For example, the deletion of 32 WO-coils, leaving just 21 along with the 12 R-coils, retains all the molecular interactions seen in the longer fibre but with fewer stacked tyrosine interactions. This can help explain the heat stability of the INP multimers and the minimal difference (2–3 °C) in activity loss between full-length *Pb*INP with 65 coils and the truncated version with 33 coils (*Figure 6*). Similarly, longer WO-coil regions can be accommodated by lengthening the gap. This can explain the wide range of WO-coil lengths seen in nature (*Figure 2*). They all fit in the same model.

Our model also shows how the interaction between the R-coils and the WO-coils of the adjacent dimers supports fibre formation. Any shortening of the R-coil subdomain jeopardizes the ability to link up the dimers. The catastrophic loss of ice nucleation activity seen below eight R-coils is because the interacting length of R- and WO-coils has too few electrostatic and other interactions to bridge the dimers together. The importance of electrostatic interaction has been illustrated in this study in two ways. First, when the R-coil basic residues were replaced by acidic residues, the ice nucleation activity was severely compromised but was fully restored when the mutated residues were all converted to lysines. Second, in cell-free extracts of lysed INP-producing *E. coli*, ice nucleation activity decreased by a few degrees Celsius at low pH values where the charge on acidic residue side chains was reduced or eliminated. When the carboxyl groups of aspartate and glutamate involved in electrostatic pairing lose their negative charges at low pH, they can still form hydrogen bonds with basic amino acid partners, which can explain why the lowering of pH was not as disruptive as reversing the charge on these residues. Another useful test of the electrostatic component to the multimer model would be to study the effects of increasing salt concentration on ice nucleation activity of the *E. coli* extracts.

The observation that low, variable levels of ice nucleation activity remained in the construct where the R-coil basic residues were replaced by acidic residues, suggests that there are additional binding interactions between the dimers other than electrostatic ones. We suggest the involvement of the highly conserved C-terminal capping structure. When three mutations designed to disrupt the cap fold were introduced, all ice nucleation activity was lost. Also of note is the disruptive effect of extending the tyrosine ladder further into the R-coil subdomain in the mutant where the acidic residues replaced the basic ones. The subtle details of the R-coil region will require detailed structural analysis for their elucidation.

The relocation of the R-coils to the N-terminal end of the solenoid caused a loss of just over 50% activity and it is possible to accommodate such a change in the model while retaining a charge interaction between the R- and WO-coils. However, the separation from the cap structure might account for some of the activity loss. Movement of the R-coils to the centre of the WO-coil region is not compatible with the model and sure enough, this construct was devoid of ice nucleation activity (*Figure 3B*).

Other features supporting the model are that the dimer's C- and N-terminal ends are exposed and can accommodate tags and extensions without disrupting the fibre. Thus, the addition of a C-terminal GFP tag has no detrimental effect in ice nucleation activity. Nor is there any difference in activity if the N-terminal INP domain and linker region are present or not (*Figure 7*; *Forbes et al., 2022*; *Kassmann-huber et al., 2020*). Even the incorporation of a bulky protein like mRuby into the WO-coil region (*Forbes et al., 2022*) can be accommodated because the fibre is just a dimer rather than a bundle of solenoids.

Electron microscopy of newly synthesized INPs in a cell-free system shows them as thin molecules of dimensions 4–6 nm in diameter by a few hundred nm in length (*Novikova et al., 2018*). Negatively stained images of recombinantly produced INP multimers isolated by centrifugation and chromatography show an elongated structure ~5–7 times longer than a monomer but not much wider (*Hartmann et al., 2022*). The fibres seen *in situ* in INP-expressing *E. coli* (*Figure 8*) are similarly long but slightly thinner, consistent with the absence of negative staining. The model in *Figure 9* is the thinnest structure we can project for a fibrillar multimer.

Solving the structure of the INP fibres at atomic detail will be the key to understanding the remarkable ability of biological ice nucleators to start the freezing process at high sub-zero temperatures. Structures of this type offer the promise of cell-free ice nucleation for use in biotechnological and food applications where there is a need to avoid the use of bacteria.

## Methods

### Materials availability statement

Newly designed genes are all modifications of the INP found in *Pseudomonas borealis* (*Pb*INP) (GenBank accession: EU573998). Their design and construction are described below.

### AlphaFold prediction

The AlphaFold model for *Pb*INP was generated by Forbes et al. as described (*Forbes et al., 2022*).

### Bioinformatic analysis of INPs

NCBI's BLAST was accessed using the BioPython library v1.81 (*Cock et al., 2009*). The consensus sequence for the 16-residue coil 'AGYGSTQTAGEDSSLT' was used as the query against the non-redundant protein database. The PAM30 scoring matrix was used due to the short query.

Quality control (QC) was performed using a custom Python script, making use of BioPython's Entrez module to fetch information on the protein, BioProject, and assembly method for each BLAST result (*Figure 1—figure supplement 2*). This script was written and run in Google Colab and has been deposited on GitHub (copy archived at *Hansen, 2023*). The same Python script was used to automatically identify the tandem repeats and classify them as WO- or R-coils.

Sequence logos were made using the Logomaker package v0.8 (*Tareen and Kinney, 2019*). Alignment of C-terminal cap sequences was performed using JalView software v2.11.2.6 (*Waterhouse et al., 2009*).

### Synthesis of *Pb*INP genes

Experiments for this project used a synthetic *Pb*INP gene previously developed by our group. This codon-optimized gene encodes the *P. borealis* INP gene (GenBank accession: EU573998). Additionally, the DNA sequence for enhanced green fluorescent protein (eGFP) (GenBank accession: AAB02572) was fused to the 3'-end of the *Pb*INP gene using a hexanucleotide encoding two linker residues (Asn–Ser). More details about the *Pb*INP-eGFP sequence are provided in *Forbes et al., 2022*.

All mutants for this study were designed by modifying the aforementioned synthetic gene. GenScript (Piscataway, NJ, USA) performed all gene syntheses, which we subsequently cloned into the pET-24a expression vector.

Five *Pb*INP mutants were designed to test the effect of replacing the R-coils. The R-coils were incrementally replaced with the sequences of WO-coils adjacent to the R-coil region as indicated, resulting in constructs containing 10, 8, 6, 4, and 1 R-coil(s) (*Figure 2A*). Replacements were designed such that they maintained the periodicity of the tandem repeats. The C-terminal R-coil was left untouched to avoid disturbing possible interactions with the putative C-terminal cap structure.

The R-coils were either relocated within the protein or deleted (*Figure 3A*), while again leaving the N- and C-terminal coils untouched to avoid interactions with adjacent domains.

Targeted mutations were introduced to the R-coil region gene to produce four additional constructs (*Figure 4A*). For the first construct ('RKH replace' in *Figure 4A*), any positively charged residues (Arg, Lys, and His) in positions 11, 12, or 14 in the R-coils were replaced with residues commonly found in those locations in the WO-coils (Asp, Glu, and Gly for positions 11 and 12, Ser for position 14). The second construct ('Y-extend' in *Figure 4A*) extends the stacked tyrosine ladder present at position 3 of the coils through seven additional coils towards the C terminus of the solenoid. The third construct ('RKH + Y-extend' in *Figure 4A*) is a combination of both mutants. The fourth construct ('K-coils' in *Figure 4A*) converted every Arg residue in the R-coil section to a Lys residue.

## Protein expression in *E. coli*

Each *Pb*INP construct was transformed into the ArcticExpress strain of *E. coli* (Agilent Technologies, Catalog #230192), since its expression of two cold-adapted chaperones, Cpn10 and Cpn60, promotes the correct folding of proteins at low temperatures (*Belval et al., 2015*). Transformation and induction with isopropyl ß-D-1-thiogalactopyranoside (IPTG) were performed according to the supplier's instructions. Cells expressed at 10 °C for 24 hr post-induction. The eGFP tag allowed expression to be confirmed using fluorescence microscopy (*Forbes et al., 2022*).

## Ice nucleation assays by WISDOM

Constructs were assayed on WISDOM (Weizmann Supercooled Droplets Observation on a Microarray) (*Reicher et al., 2018*) in a similar way as described in *Forbes et al., 2022*.

## Ice nucleation assays by nanoliter osmometer

Ice nucleation activity was quantified using a droplet freezing assay protocol (*Lee et al., 2023*) that makes use of a LabVIEW-operated nanoliter osmometer (Micro-Ice, Israel) (*Braslavsky and Drori, 2013*). Briefly: Following induction and cold incubation, nanoliter-sized droplets of liquid cultures were pipetted into oil-filled wells resting on a cold stage. The temperature of the cold stage was lowered at a rate of 1 °C/min while a video recording was taken of the sample grid. Freezing was characterized by a distinct change in droplet appearance. After assay completion, the videos were analysed to record the temperatures of all freezing events. The fraction of frozen droplets ($f_{ice}$) as a function of temperature was plotted, generating ice nucleation curves for each sample. This apparatus could not reach temperatures as low as those achieved on WISDOM, but results are in agreement between the two approaches (*Figure 2B*).

## Heat treatment and pH

To obtain cell lysates, *E. coli* cultures were centrifuged at 3200 × *g* for 30 min post-induction. Cell pellets were then resuspended in a lysis buffer of 50 mM Tris–HCl, 150 mM NaCl, containing Pierce Protease Inhibitor (Thermo Scientific, Canada) before sonication at 70% amplitude for 30 s rounds. Lysate was centrifuged at 31,000 × *g* and the resulting supernatant was passed through a 0.2-μm filter.

For heat treatment, filtered lysate in sealed Eppendorf tubes was heated at 60, 70, 80, 90, or 99 °C for 10 min in a thermocycler and then quenched on ice prior to being assayed for activity.

For the pH experiments, aliquots of filtered lysate were diluted 50-fold in pH-adjusted buffer of 100 mM sodium citrate, 100 mM sodium phosphate, and 100 mM sodium borate following the protocol by *Chao et al., 1994*. Before assaying, we verified using universal indicator strips that addition of lysate to the buffer mixtures did not meaningfully affect the pH of the final mixtures.

## Preparation of the cryo-EM grids

After confirming eGFP-INP expression, the *E. coli* cultures were incubated at 4 °C for an additional 3 days. The *E. coli* cells were spun down and resuspended in PBS to an OD$_{600}$ of ~3. These concentrated

*E. coli* samples were deposited onto freshly glow-discharged QUANTIFOIL holey carbon grids (Electron Microscopy Sciences). The grids were then blotted from the back side with the filter paper for 5 s before being plunge frozen in liquid ethane, using a manual plunger-freezing apparatus as described previously (*Liu et al., 2009*; *Zhao et al., 2013*).

### Cryo-FIB milling

The plunge-frozen grids with *E. coli* cells were clipped into cryo-FIB AutoGrids and mounted into the specimen shuttle under liquid nitrogen. An Aquilos2 cryo-FIB system (Thermo Fisher Scientific) was used to mill the thick bacterial samples into lamellae of <200 nm in thickness. The milling process was completed using a protocol as previously described (*Xiang et al., 2021*).

### Cryo-ET data acquisition and tomogram reconstruction

Grids containing the lamellae obtained from cryo-FIB milling were loaded into either a 300-kV Titan Krios electron microscope (Thermo Fisher Scientific) equipped with a Direct Electron Detector and energy filter (Gatan) or a 200-kV Glacios Electron Microscope at Yale University. The FastTOMO script was used with the SerialEM software to collect tilt series with defocus values of approximately −6 μm (*Xu and Xu, 2021*), and a cumulative dose of 70 e⁻/Å covering angles from −48° to 48° (3° tilt step). Images were acquired at ×42,000 magnification with an effective pixel size of 2.148 Å. All recorded images were first drift corrected by MotionCor2 (*Zheng et al., 2017*), stacked by the software package IMOD (*Kremer et al., 1996*), and then aligned by IMOD using Pt particles as fiducial markers. TOMO3D was used to generate tomograms by simultaneous iterative reconstruction technique (SIRT) (*Agulleiro and Fernandez, 2015*). In total, 10 tomograms were reconstructed with TOMO3D for the wild-type INP while 5 tomograms were produced for the R-coil mutant.

### Statistics

p-values reported were calculated by one-way analysis of variance.

## Acknowledgements

This work was supported by CIHR Foundation Grant (FRN 148422) to PLD, who holds the Canada Research Chair in Protein Engineering, and by an Israel Science Foundation grant to IB. YR acknowledges support by a research grant from the Yotam project and the Weizmann Institute Sustainability and Energy Research Initiative. SG was supported by a CIHR Post-Doctoral Fellowship and an NIH RO1 grant (R01AI087946) of JL. WG was also supported by the NIH RO1 grant (R01AI087946) of JL. We thank Virginia K Walker for the gift of the *Pseudomonas borealis* strain.

## Additional information

#### Competing interests

Ido Braslavsky: Ido Braslavsky is a co-founder of Micro-ice. The other authors declare that no competing interests exist.

#### Funding

| Funder | Grant reference number | Author |
| --- | --- | --- |
| Canadian Institutes of Health Research | FRN 148422 | Peter L Davies |
| Israel Science Foundation | | Ido Braslavsky |
| Weizmann Institute of Science | | Yinon Rudich |
| Canadian Institutes of Health Research | Post-Doctoral Fellowship | Shuaiqi Guo |
| National Institute of Allergy and Infectious Diseases | R01AI087946 | Jun Liu<br>Wangbiao Guo |

| Funder | Grant reference number | Author |
|--------|------------------------|--------|

The funders had no role in study design, data collection, and interpretation, or the decision to submit the work for publication.

## Author contributions

Thomas Hansen, Conceptualization, Data curation, Software, Formal analysis, Investigation, Visualization, Methodology, Writing – original draft, Writing – review and editing; Jocelyn Lee, Conceptualization, Formal analysis, Methodology, Writing – original draft, Writing – review and editing; Naama Reicher, Formal analysis, Validation, Investigation, Methodology; Gil Ovadia, Wangbiao Guo, Investigation; Shuaiqi Guo, Formal analysis, Investigation, Visualization, Methodology, Writing – original draft, Writing – review and editing; Jun Liu, Resources, Supervision, Funding acquisition; Ido Braslavsky, Resources, Supervision, Funding acquisition, Project administration, Writing – review and editing; Yinon Rudich, Resources, Supervision, Funding acquisition, Methodology, Project administration, Writing – review and editing; Peter L Davies, Conceptualization, Resources, Formal analysis, Supervision, Funding acquisition, Methodology, Writing – original draft, Project administration, Writing – review and editing

## Author ORCIDs

Thomas Hansen ⓘ https://orcid.org/0000-0001-7515-9540
Jocelyn Lee ⓘ https://orcid.org/0009-0005-4431-5439
Jun Liu ⓘ http://orcid.org/0000-0003-3108-6735
Peter L Davies ⓘ https://orcid.org/0000-0002-8026-7818

Reviewer #1 (Public Review): https://doi.org/10.7554/eLife.91976.3.sa1
Reviewer #2 (Public Review): https://doi.org/10.7554/eLife.91976.3.sa2
Reviewer #3 (Public Review): https://doi.org/10.7554/eLife.91976.3.sa3
Author Response https://doi.org/10.7554/eLife.91976.3.sa4

# Additional files

## Supplementary files

- Source data 1. Droplet freezing assay results.
- MDAR checklist

## Data availability

Source data are provided for the bioinformatic analysis (*Figure 1—source data 1*) as well as for droplet freezing assays (Figures 2–7: *Source data 1*).

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
