## [Editor Report · eLife assessment]

This **valuable** study provides molecular-level insights into the functional mechanism of bacterial ice-nucleating proteins, detailing electrostatic interactions in the domain architecture of multimeric assemblies. The evidence supporting the claims of the authors is **solid**, with results from protein engineering experiments, functional assays, and cryo-electron tomography, while the proposed structural model of protein self-assembly remains hypothetical. The work is of broad interest to researchers in the fields of protein structural biology, biochemistry, and biophysics, with implications in microbial ecology and atmospheric glaciation.

---

## [Referee Report · Reviewer #1 (Public Review)]

Summary: Hansen et al. dissect the molecular mechanisms of bacterial ice nucleating proteins mutating the protein systematically. They assay the ice nucleating ability for variants changing the R-coils as well as the coil capping motifs. The ice nucleation mechanism depends on the integrity of the R-coils, without which the multimerization and formation of fibrils are disrupted.

Strengths: The effects of mutations are really dramatic, so there is no doubt about the effect. The variants tested are logical and progressively advance the story. The authors identify an underlying mechanism involving multimerization, which is plausible and compatible with EM data. The model is further shown to work in cells by tomography.

Weaknesses: The theoretical model presented for how the proteins assemble into fibrils is simple, but not supported by much data.

---

## [Referee Report · Reviewer #2 (Public Review)]

Summary:

This paper further investigates the role of self-assembly of ice-binding bacterial proteins in promoting ice-nucleation. For the P. borealis Ice Nucleating Protein (PbINP) studied here, earlier work had already determined clearly distinct roles for different subdomains of the protein in determining activity. Key players are the water-organizing loops (WO-loops) of the central beta-solenoid structure and a set of non-water-organizing C-terminal loops, called the R-loops in view of characteristically located arginines. Previous mutation studies (using nucleation activity as a read-out) had already suggested the R-loops interact with the WO loops, to cause self-assembly of PbINP, which in turn was thought to lead to enhanced ice-nucleating activity. In this paper, the activities of additional mutants are studied, and a bioinformatics analysis on the statistics of the number of WO- and R-loops is presented for a wide range of bacterial ice-nucleating proteins, and additional electron-microscopy results are presented on fibrils formed by the non-mutated PbINP in E coli lysates.

Strengths:

-A very complete set of additional mutants is investigated to further strengthen the earlier hypothesis.

-A nice bioinformatics analysis that underscores that the hypothesis should apply not only to PbINP but to a wide range of (related) bacterial ice-nucleating proteins.

-Convincing data that PbINP overexpressed in E coli forms fibrils (electron microscopy on E coli lysates).

Weaknesses:

-The new data is interesting and further strengthens the hypotheses put forward in the earlier work. However, just as in the earlier work, the proof for the link between self-assembly and ice-nucleation remains indirect. Assembly into fibrils is shown for E coli lysates expressing non-mutated pbINP, hence it is indeed clear that pbINP self-associates. It is not shown however that the mutations that lead to loss of ice-nucleating activity also lead to loss of self-assembly. A more quantitative or additional self-assembly assay could shine light on this, either in the present or in future studies.

-Also the "working model" for the self-assembly of the fibers remains not more than that, just as in the earlier papers, since the mutation-activity relationship does not contain enough information to build a good structural model. Again, a better model would require different kinds of experiments, that yield more detailed structural data on the fibrils.

---

## [Referee Report · Reviewer #3 (Public Review)]

Summary: in this manuscript, Hansen and co-authors investigated the role of R-coils in the multimerization and ice nucleation activity of PbINP, an ice nucleation protein identified in Pseudomonas borealis. The results of this work suggest that the length, localization, and amino acid composition of R-coils are crucial for the formation of PbINP multimers.

Strengths: The authors use a rational mutagenesis approach to identify the role of the length, localisation, and amino acid composition of R-coils in ice nucleation activity. Based on these results, the authors hypothesize a multimerization model. Overall, this is a multidisciplinary work that provides new insights into the molecular mechanisms underlying ice nucleation activity.

Weaknesses: Several parts of the work appear cryptic and unsuitable for non-expert readers. The results of this work should be better described and presented.

---

## [Author Response]

The following is the authors’ response to the original reviews.

**Public Reviews:**

**Reviewer #1 (Public Review):**
Summary: Hansen et al. dissect the molecular mechanisms of bacterial ice nucleating proteins mutating the protein systematically. They assay the ice nucleating ability for variants changing the R-coils as well as the coil capping motifs. The ice nucleation mechanism depends on the integrity of the R-coils, without which the multimerization and formation of fibrils are disrupted.Strengths: The effects of mutations are really dramatic, so there is no doubt about the effect. The variants tested are logical and progressively advance the story. The authors identify an underlying mechanism involving multimerization, which is plausible and compatible with EM data. The model is further shown to work in cells by tomography.Weaknesses: The theoretical model presented for how the proteins assemble into fibrils is simple, but not supported by much data.

Agreed. This theoretical INP multimer model was introduced to promote discussion and elicit ideas on how to prove or disprove it. The length and width of the fibres are defined by cryo-ET results, in which the narrow width is just sufficient to accommodate a dimer of the INPs, and the long length requires that several INPs are joined end to end. Their antiparallel arrangement produces identical ends to the dimer and avoids steric clash of the C-terminal cap structures as well as the C-terminal GFP tag. This model can accommodate the wide range of INPs lengths seen in nature (due to different numbers of water-organizing coils) and introduced in mutagenesis experiments (Forbes et al. 2022). It defines a critical role for the R-coil subdomain in joining the dimers together and explains why this region cannot be shortened by more than a few coils either in nature or by experimentation.

In response to specific criticisms of the model (Fig. 9), we have redesigned this to be less schematic and to incorporate several copies of the AlphaFold-predicted structure.

**Reviewer #2 (Public Review):**
Summary:This paper further investigates the role of self-assembly of ice-binding bacterial proteins in promoting ice-nucleation. For the P. borealis Ice Nucleating Protein (PbINP) studied here, earlier work had already determined clearly distinct roles for different subdomains of the protein in determining activity. Key players are the water-organizing loops (WO-loops) of the central beta-solenoid structure and a set of non-water-organizing C-terminal loops, called the R-loops in view of characteristically located arginines. Previous mutation studies (using nucleation activity as a read-out) had already suggested the R-loops interact with the WO loops, to cause self-assembly of PbINP, which in turn was thought to lead to enhanced ice-nucleating activity. In this paper, the activities of additional mutants are studied, and a bioinformatics analysis on the statistics of the number of WO- and R-loops is presented for a wide range of bacterial ice-nucleating proteins, and additional electron-microscopy results are presented on fibrils formed by the non-mutated PbINP in E coli lysates.Strengths:-A very complete set of additional mutants is investigated to further strengthen the earlier hypothesis.-A nice bioinformatics analysis that underscores that the hypothesis should apply not only to PbINP but to a wide range of (related) bacterial ice-nucleating proteins.-Convincing data that PbINP overexpressed in E coli forms fibrils (electron microscopy on E coli lysates).Weaknesses:-The new data is interesting and further strengthens the hypotheses put forward in the earlier work. However, just as in the earlier work, the proof for the link between self-assembly and ice-nucleation remains indirect. Assembly into fibrils is shown for E coli lysates expressing non-mutated pbINP, hence it is indeed clear that pbINP self-associates. It is not shown however that the mutations that lead to loss of ice-nucleating activity also lead to loss of self-assembly. A more quantitative or additional self-assembly assay could shine light on this, either in the present or in future studies.

The control cryo-ET experiment where the R-coils were deleted and INP fibres were not seen is consistent with a link between the loss of ice-nucleating activity and the loss of self-assembly. However, we agree that a more direct measurement of the physical state of INP molecules is needed to prove the link.

-Also the "working model" for the self-assembly of the fibers remains not more than that, just as in the earlier papers, since the mutation-activity relationship does not contain enough information to build a good structural model. Again, a better model would require different kinds of experiments, that yield more detailed structural data on the fibrils.

Reviewer #1 also raised these criticisms of the model, which we have responded to (above). Testing the model is a focus of our continuing experiments on INPs.

**Reviewer #3 (Public Review):**
Summary: in this manuscript, Hansen and co-authors investigated the role of R-coils in the multimerization and ice nucleation activity of PbINP, an ice nucleation protein identified in Pseudomonas borealis. The results of this work suggest that the length, localization, and amino acid composition of R-coils are crucial for the formation of PbINP multimers.Strengths: The authors use a rational mutagenesis approach to identify the role of the length, localisation, and amino acid composition of R-coils in ice nucleation activity. Based on these results, the authors hypothesize a multimerization model. Overall, this is a multidisciplinary work that provides new insights into the molecular mechanisms underlying ice nucleation activity.Weaknesses: Several parts of the work appear cryptic and unsuitable for non-expert readers. The results of this work should be better described and presented.

In revising the manuscript for reposting we have rewritten sections to make it more accessible to the non-expert. Incorporating the detailed recommendations of the reviewers has been helpful in this effort.

Recommendations for the authors: please note that you control which revisions to undertake from the public reviews and recommendations for the authors

**Reviewer #1 (Recommendations For The Authors):**
Introduction: Curiously, there is no mention at all in the introduction of what the biological function of these ice-nucleating proteins is.

We added the following text to the first paragraph of the Introduction: ”INP-producing bacteria are widespread in the environment where they are responsible for initiating frost (4) and atmospheric precipitation (5). As such, these bacteria play a significant role in the Earth’s hydrological cycle and in agricultural productivity.”

Line 70: TXT, SLT, and Y motifs are mentioned, but only the first is described. Also, TXT name alternates between TXT and TxT in the manuscript. (I think the latter is more correct).

These putative water-organizing motifs are introduced in the preceding paper (new ref 8). We now use TxT consistently throughout the manuscript and have converted SLT to SxT because L is an inward-pointing residue that is not directly involved in water organization.

Line 236: A construct with repeats deleted is tested for thermostability, but it is not really explained what hypothesis this experiment is supposed to test.

This is an observation that adds information about the stability of the INP multimers and will need to be explained by the structure.

Line 267: The authors test a mutant where the N-terminal coil is disrupted and find a big effect. Nevertheless, no conclusion is drawn. What does this result mean?

On the contrary, INP activity is not appreciably affected by N-terminal deletion.

Line 269: The CryoEM begins rather abruptly with technical details. Consider introducing the paragraph with a brief statement about what you want to investigate. Also, the analysis seems a little half-hearted.Given that the authors describe other EM studies of fibrils of the same protein it would be nice with a clear statement about what is new in their study and how it compares to previous studies.

We have added this statement about why we used Cryo-EM: “The idea that INPs must assemble into larger structures to be effective at ice nucleation has persisted since their discovery (6). In the interim the resolving power of cryo-EM has immensely improved. Here we elected to use cryo-electron tomography to view the INP multimers *in situ* and avoid any perturbation of their superstructure during isolation.”

Fig. 7B: Single-letter amino acid codes are always capitalized.

We have revised this figure to use capital letters for the amino acids.

Fig. 9: This figure is really hard to read even though it is very simplistic. I would consider making a figure with several copies of the AlphaFold model instead. Especially panel D, I do not know what is supposed to show.

We have followed this advice and have completely revised the figure using copies of the AlphaFold model. Panel D (now C) shows two cross-sections through the AlphaFold model.

Line 355 onwards: The model of the INP is the weakest part of the manuscript. This reviewer considers that the model is crude and it is unclear what information the model is supported by. The authors might want to consider running an AlphaFold multimer to get a better model of at least the dimer.

Our objective now is to validate or disprove the model by experimentation using protein-protein cross-linking in conjunction with mass spectrometry, and higher resolution cryo-EM methods.

**Reviewer #2 (Recommendations For The Authors):**
I would suggest more frankly discussing the weaknesses mentioned in my public review, as well as approaches that could be used in the future to address these.

In the cryo-ET analysis, INP mutations of the R-coils that lead to loss of ice-nucleating activity fail to show fibres in the bacteria (Fig. S4), which is consistent with the loss of self-assembly. We are working on physical methods that can assess the degree of assembly of the different INP constructs and mutations.We are working to validate and improve the working model of INP multimers.

**Reviewer #3 (Recommendations For The Authors):**
AbstractLine 18. Below 0 Celsius should be < 0 {degree sign}C.

Done

Line 25. *E. coli* should be *Escherichia coli*

Done

Line 29. *E. coli* should be in italics.

Done

IntroductionThe introduction is weak and not suitable for non-expert readers. Moreover, in some parts it is cryptic and it is not clear whether the authors are describing INP in general or PbINP. The introduction should be reorganized to highlight the novelty of this paper compared to Forbes et al. 2022.

The changes we have made to the Introduction can be seen in the ‘documents compared’ version where the changes are tracked.

Line 45. It is unclear whether this paragraph is a result reported in the literature or the result of this work. Please clarify.

These are results reported in the literature as indicated by the references cited in the paragraph.

Line 54. It is not clear whether this paragraph describes PbINP or INP in general.

This paragraph begins with INPs in general and then focuses on PbINP.

ResultsLine 109. This section would benefit from a paragraph in which the authors describe the rationale for this bioinformatic analysis.

We added the following Statement: “A bioinformatic analysis of bacterial INPs was undertaken to identify their variations in size and sequence to understand what is common to all that could guide experiments to probe higher order structure and help develop a collective model of the INP multimer.”

Some information is needed on the selected sequences such as sequence identity, what do the authors mean by nr database?

The abbreviation nr has been replaced by ‘non-redundant’. As explained in that same paragraph the sequences selected were those from long-read sequences that could be relied on to accurately count the number of solenoid coils.

Line 144. The standard deviation is necessary to understand whether these differences are statistically significant.

These have been added as p values.

Figure 2. I noticed that the authors used GFP-tagged PbINP. Why? In addition, panel C is never mentioned in the manuscript.

The GFP tag was used to confirm expression of the PbINP in *E. coli*. We have added this sentence: “As previously described these constructs were tagged with GFP as an internal control for INP production, and its addition had no measured effect on ice nucleation activity (8).”The GFP tag was also useful as in internal control for the heat denaturation experiments featured in Fig. 6, where it lost its fluorescence between 65 and 75 °C.

Fig. 2C is now cited alongside Fig. 2B.

Figure 3. In my opinion, the results of the R-coil deletion should also be shown in Figure 2.Line 171. This section is cryptic. A logo sequence or an alignment of WO-coils and R-coils of PbINP could be helpful for the reader. Instead of the architecture of the whole protein, it would be useful to have the sequence of the R-coils with the residues that the authors mutagenised.

The logo sequences are available in Fig. 1.

Line 202. Here, the authors describe a new experimental setup. As the Materials and Methods section follows the Discussion, the authors should state in the first paragraph of the Results section that IN activity was measured on whole cells.

We have now modified the introductory sentence to read: “Ice nucleation assays were performed on intact *E. coli* expressing PbINP to assess the activity of the incremental replacement mutants.”

Line 202. The authors investigated the effects of pH and temperature (Line 223) on the IN activity. The authors should better introduce the rationale for these experiments and how they fit within the work.

We have now modified the following sentence to provide the rationale: “To see how important electrostatic interactions were in the multimerization of PbINP as reflected by its ice nucleation activity, it was necessary to lyse the *E. coli* to change the pH surrounding the INP multimers.”

Line 245. This work is supported by a model provided by Alphafold. I wonder how reliable this model is; the authors should indicate the quality of the model and provide the accuracy values of the residuals.

This information is now provided in Figure S1.

Line 259. Typically in mutagenesis studies, a key residue is substituted with alanine to create a loss of function variant. In this case, the authors have made the following substitutions F1204D, D1208L, and Y1230D, it is not clear to me why the authors have replaced an aromatic residue with one of aspartic acid that is negatively charged.

We have justified these more extreme changes as follows: “For an enhanced effect of the mutations hydrophobic residues were replaced with charged ones and vice versa.”

Line 269. This paragraph seems completely unrelated to the section entitled: The β-solenoid of INPs is stabilized by a capping structure at the C terminus, but not at the N terminus.

We had omitted the sub-heading “Cryo-electron tomography reveals INPs multimers form bundled fibres in recombinant cells”, which is now in place.

DiscussionOverall, the discussion is too long and some parts appear cryptic, this section should be reorganized.

The changes we have made to the Discussion can be seen in the ‘documents compared’ version where the changes are tracked.

Line 354. It is not clear what experimental evidence supports this model. In the results, this model is never mentioned and it is not clear whether it was obtained by computational analysis or not.

The model is presented in the Discussion because it was not arrived at by experimentation but is an attempt to integrate the observations made in the Results section. The experimental evidence that supports this model is reviewed in the Discussion section: “Working model of the INP multimer is consistent with the properties of INPs and their multimers.”

Line 354. The authors used GFP-tagged PbINP. The Authors should discuss the role of GFP in this model and IN activity. A measurement of IN activity on PbINP without GFP would be useful.

We have previously shown in Ref 8 that the GFP tag has no detrimental effect on ice nucleation activity. Our model for the INP multimer can accommodate this C-terminal tag without any steric hindrance.

Line 364. The Authors hypothesize that electrostatic interactions stabilize end-to-end dimer associations. To test this hypothesis, the authors should measure the activity of IN at increasing concentrations of NaCl. It is known that high salt concentrations shield charges by preventing the formation of electrostatic intermolecular interactions.

We have added this sentence to the Discussion: “Another useful test of the electrostatic component to the multimer model would be to study the effects of increasing salt concentration on ice nucleation activity of the *E. coli* extracts.”

Line 439. Conclusions should be useful for the reader.Material and MethodsIn several sections, the authors refer to what has already been published in Forbes et al. However, the minimum information should also be described in this work. In addition, the Authors should indicate the number of replicates.

The ice nucleation assays on whole cells were done on the WISDOM apparatus, which integrates 100’s of individual measurements to obtain a T50 value. These T50 values were confirmed by assays on the nanoliter osmometer apparatus. The numbers of replicates used on the nanoliter osmometer apparatus are indicated by box and whisker plots in Figs. 5 & 6 with boxes and bars showing quartiles, with medians indicated by a centre line.

Line 500. This paragraph should be removed as the results are not described in the manuscript.

This is a Methods section that describes how that INPs were expression in *E. coli*. It has details that are important for researchers who want to repeat our findings, such as the use of the Arctic Express strain for producing INP.